# Analysis of Building Accessibility Using Inertial and Optical Sensors

**DOI:** 10.3390/s23125491

**Published:** 2023-06-10

**Authors:** Tomás E. Martínez-Chao, Agustín Menéndez-Díaz, Silverio García-Cortés, Pierpaolo D’Agostino

**Affiliations:** 1Department of Civil, Building and Environmental Engineering, University of Naples “Federico II”, 80125 Naples, Italy; tomasenrique.martinezchao@unina.it (T.E.M.-C.); pierpaolo.dagostino@unina.it (P.D.); 2Department of Construction and Manufacturing Engineering, Oviedo of University, 33004 Oviedo, Spain; 3Department of Mining Exploitation and Prospecting, University of Oviedo, 33004 Oviedo, Spain; sgcortes@uniovi.es

**Keywords:** optical sensors, inertial sensors, accessibility, wheelchair, optimal routes

## Abstract

The inclusive use of urban spaces necessitates detailed knowledge of the accessibility of public buildings or places where educational, health or administrative services are provided. Despite the improvements already made in many cities regarding architectural work, further changes to public buildings and other spaces, such as old buildings or areas of historical importance, are still required. To study this problem, we developed a model based on photogrammetric techniques and the use of inertial and optical sensors. The model allowed us to perform a detailed analysis of urban routes in the surroundings of an administrative building, by means of mathematical analysis of pedestrian routes. It was applied to the specific case of people with reduced mobility and included analysis of the building accessibility as well as detection of suitable transit routes, deterioration of the road surfaces and the presence of architectural obstacles on the route.

## 1. Introduction

Universal accessibility is a right for everyone and, therefore, a fundamental condition to consider in the development of built urban spaces [1,2]. Great effort has been made since the 1980s to remove architectural barriers and material obstacles that prevent people with disabilities from using urban spaces [3]. Laws and regulations have been dictated in favour of safe and inclusive accessibility to common spaces and public and private buildings, such as the Americans with Disabilities Act (ADA) [4], the Disability Discrimination Act issued in the United Kingdom in 1995 [5], the Disability Discrimination Act issued in Australia in 1992 [6] and the General Law on Disability or LGD (former LISMI) issued in Spain in 2013 [7]. Studies have also been carried out to favour inclusivity when designing space, buildings or entire cities [8]. Other developments of great scope and importance have also been made, including the ambitious proposal of the 193 member states of the United Nations for Sustainable Development, framed in an agenda with 17 goals and 169 targets to be met by 2030. This proposal is based on five fundamental dimensions: people, prosperity, planet, collective participation and peace [9]. Specifically, goal 11 (to make cities and human settlements inclusive, safe, resilient and sustainable) recognizes the right of people with disabilities to enjoy full social participation [10] without being discriminated against, with mobility as a fundamental factor in such participation [11]. Ensuring safety and providing information on urban routes [12] contribute to achieving these goals. Despite the extensive work that has been carried out, there are still significant gaps between the constructive design of urban environments and the needs of people with disabilities [13,14].

The systematic control of existing road infrastructures is also considered in order to prolong the useful life and improve the management of these infrastructures [15]. Several factors can deteriorate road surfaces (e.g., prolonged use, poor management of the materials necessary for their construction and inclement weather [16]), in some cases even creating architectural barriers to people with mobility problems. Informative or advertising signs and decorative objects can benefit many people, but their inadequate management may also hinder the movement of others.

New remote sensing and geoinformation technologies have opened up innovative scenarios in several fields of information and urban mobility [17]. The use of survey equipment, such as unmanned aerial vehicles (UAV) [18] and laser scanners, provides accurate data on the studied area and also speed up data acquisition and processing, allowing the analysis of the areas in near real-time [19].

The different types of architectural barriers that exist must be analysed in order to determine accessibility before designing pedestrian routes [20]. Urban barriers (in public spaces and roads) are among the main barriers generally found outside buildings [21,22,23,24], including streets that are too narrow for wheelchair users [21,23,25,26], stairs that are not adapted, kerbs that do not have ramps to allow passage between the pavement and the roadway, ramps with steeper slopes or that are narrower than permitted, obstacles in the middle of pavements, such as road/advertising signs, traffic lights, lampposts, etc., and also the usual traffic flow [22,25,27] (Figure 1), all of which restrict the normal movement of people. These types of barriers are often found in older urban areas, but also in cities where there is a lack of management and control over new infrastructures to ensure the minimum characteristics of walkability and inclusiveness (Figure 2).

Other types of architectural barriers in public buildings prevent people with physical disabilities having proper access [28]. These include the lack of ramps, escalators and lifts, and the lack of adequate signs for people with visual disabilities. Similar barriers in private homes may include the lack of ramps and lifts or doors that are too narrow to allow wheelchair access. Barriers in public transport [29,30,31] include lack of ramps, escalators or lifts in train or metro stations and the lack of space for wheelchairs on buses. Barriers in historic or cultural heritage buildings may include old architecture of the building or lack of adaptation measures to allow access for people with disabilities. Another important aspect to consider in pedestrian routes is the level of deterioration of the route, as well as the type of surface. When a wheelchair moves over a deteriorated surface, the vibrations generated can be quite perceptible and uncomfortable for the users [32,33,34,35]. Some common vibrations are:High-frequency vibrations caused by small bumps or cracks in the pavement. These can produce a buzzing sensation in the hands, arms and body of the wheelchair user;Low-frequency vibrations caused by larger cracks or uneven surfaces. These produce a slower and more pronounced shaking sensation throughout the body;Lateral vibrations, which occur when the wheelchair moves over an uneven surface, causing the wheels to wobble from side to side. These are particularly uncomfortable, as the user may feel that the wheelchair is tipping over;Vertical vibrations, which occur when the wheelchair hits a bump or a steep slope in the pavement, causing the user to experience a sudden jolt or impact. This is especially uncomfortable for people with back pain or other mobility problems.

The main objective of this research was to automate the analysis and decision-making involved in characterizing an urban itinerary and establishing the level of accessibility, mainly focusing on people with reduced mobility, such as wheelchair users. The methodology goes beyond a preliminary, static study of accessibility conditions, in the sense of incorporating dynamic sensors for real-time evaluation of the deterioration of architectural elements. We conducted an analysis to determine the real conditions of accessibility of an urban itinerary, by using data acquisition instruments. The data obtained allowed us to determine the level of accessibility for different types of people of a specific route, as well as to suggest alternative paths so that the most difficult routes can be avoided.

This paper is structured as follows. The application of the proposed methodology to a synthetic 3D building model is described in Section 2. All of the procedures performed are explained, starting from the input data to classify the accessibility of the route for a wheelchair user. The application of the methodology to a real case is reported in Section 3. Data were acquired with optical sensors (point cloud obtained from drone imagery) and used to determine the conditions and path of best route of entry to a public building. Data acquired with an inertial sensor were also considered, with the intention of improving the analysis of the conditions of the routes in relation to the comfort of wheelchair users. This was analysed from the point of view of shortest distance, adequate width, slope and vibration level, specifically for wheelchair users. The results of the analysis and the associated cost of each feasible route, aimed at selecting the best route, are presented in Section 4. The results are critically discussed in Section 5. Finally, the conclusions are presented in Section 6.

## 2. Methodology

### 2.1. Related Works

Accessibility is determined by analysing various factors related to infrastructure, transportation, services and policies that affect people’s ability to move and participate in urban spaces [36]. This, in turn, is determined by the spatial distribution of potential destinations, the ease of reaching each destination and the extent, quality and character of the activities that take place at each [37]. From a conceptual point of view, measurement or assessment of accessibility is divided into 7 main steps: (1) counting the number of locations where an activity can be accessed easily, (2) summing the spatial separations between one location and all other locations, (3) determining the nearest available location for a given service, (4) measuring accessibility to a specific type of person starting at a particular location or zone in relation to some activity, (5) establishing the maximum and net benefit of a route, (6) determining the probability that a person in the zone will travel to another zone for an activity involving a specific type of travel and (7) the absolute accessibility to a building in connection with the existence or not of insuperable barriers.

The accessibility measures mentioned above are described in further detail by [38], which also explains accessibility measures inside and around buildings. In order to analyse accessibility in the areas surrounding buildings, there are three ways of considering such areas. The first scenario concentrates on the elements comprising the area immediately outside the building, i.e., the state of the road surface, accessibility of the entrance, etc. The second refers to the parking areas (and routes) associated with the buildings. Finally, the third refers to all transport routes leading to the buildings. In the latter case, the authority in charge of managing the routes will not usually be the same as that in charge of the building itself, so the responsibility for the care and control of accessibility conditions will not fall directly on the building management. Our work concentrates on the first two scenarios, which are largely the responsibility of public building management.

Other methods can be used to measure accessibility. In addition, the methods used to acquire data and the scale of the work will vary depending on the level of detail of the analysis. Examples of other studies conducted at different scales, using different analysis and data acquisition methods are listed in Table 1.

### 2.2. General Workflow

The walkability, safety and comfort of potential routes must be assessed in order to establish the most suitable for wheelchair users. A methodology based on data acquired with unmanned aerial vehicles (UAVs) was used for this purpose. In addition, architectural plans, base maps, three-dimensional models, etc., of the areas of interest were generated and different types of analysis were carried out to extract slope, width, length parameters and the level of degradation of the routes, the latter with the help of inertial sensors.

The methodology implemented was initially analysed in a virtual set-up. As shown in Figure 3, a three-dimensional model built from the Blender software was used as the input data for an urban environment. This model was recreated on a real terrain surface, obtained through a photogrammetric process with UAV. Based on this data, determination of the optimal mobility routes was divided in 4 stages:Stage 1 (S1): analysis of spatial data for the identification and removal of the obstacles and generation of a digital elevation model (DEM);Stage 2 (S2): analysis of interaction with obstacles, barriers and building edges to identify the accessibility zone (AZ), i.e., the region of the study area that is free of obstacles, where it is possible to circulate safely;Stage 3 (S3): identification of the medial axis (MA), the departure/starting point (SP), the arrival/end point (EP), and other key points (KP). This process was performed for each zone (buildings, entrances, bus stops, access points to parking lots, etc.) to analyse the possible routes for each AZ;Stage 4 (S4): classification of MA branches and generation of the cost matrix (CM), according to the degree of difficulty in terms of mobility, safety and comfort.

### 2.3. Stage 1: Preliminary Data Processing

The 3D model designed was imported in the CloudCompare software as a mesh and transformed into a point cloud, allowing the identification of the obstacles and the regions corresponding to the ground. For this purpose, the subsets of connected components were extracted according to a heuristic criterion based on the algorithmic application of graph theory [45], with an Octree level of 12 and a minimum of 1000 points per component. The selected values minimize the noise corresponding to scattered points in the pedestrian area. The obstacles identified were classified according to their characteristics and shape, in a similar way to other procedures [33,34].

We then focused on the region identified as the ground surface. The cloth simulation filtering (CSF) algorithm [46,47], with the general relief option, was used to filter the surface points. The advanced parameters were adjusted by visual control to find the combination best adapted to the bare ground surface. The resolution and threshold were set to 0.1 m with a maximum of 500 iterations. Thus, the filtering process was divided into two parts. The first involved the use of the CSF algorithm, which returned two point clouds and one mesh. This process generated some gaps, which were defined to enable the calculation of accessibility. To correct these defects, a second filtering process was carried out in which the distance of all the points of the original cloud to the mesh generated by the CSF was calculated; at this point, the cloud-to-mesh distance algorithm was used. The purpose of this second process was to correct the small imperfections caused by the output of the CSF filtering.

We computed the distance of the points of the mesh obtained with the point cloud by a cloud-to-mesh distance algorithm, which reassigns the values of the height of the points relative to the newly calculated surface. The point cloud is filtered by the values corresponding to the surface, which is converted to a DEM raster with a mesh size of 0.01 m projected on the perpendicular axis without reassigning the values of the empty cells. The DEM raster obtained was exported to QGIS software to continue to Stage 2.

### 2.4. Stage 2: Identification of the AZ

The slope generation algorithm with GDAL was used in the QGis software to compute the slope of the pixels by means of the Horn method [48] and to classify them as flat or steep. The Horn method, also known as the method of the eight slopes, is an algorithm commonly used in the topographic analysis of raster data to determine the slope of a cell or pixel from its eight neighbours. The following formulas are used to calculate the slope of a single pixel:(1)$$S_{x}=\frac{\left(Z_{1}+2Z_{2}+Z_{3}\right)-\left(Z_{7}+2Z_{8}+Z_{9}\right)}{8c_{x}}$$
(2)$$S_{y}=\frac{\left(Z_{3}+2Z_{4}+Z_{8}\right)-\left(Z_{1}+2Z_{6}+Z_{7}\right)}{8c_{y}}$$
(3)$$S=\sqrt{S_{x}^{2}+S_{y}^{2}}$$
where *S* is the slope of the cell, *S_x_* and *S_y_* its components, *Z*_1–9_ the elevation values of the neighboring pixels, and *c_x_* and *c_y_* are the cell resolutions along each direction.

The pixels are classified as flat (steep) according to a low (high) pass filter with a maximum (minimum) slope of 10%. They are then vectorized into a layer of polygons, with values of 1 and 0 corresponding to flat or steep areas, respectively. The flat polygons are extracted as a new vector layer called AZ.

### 2.5. Stage 3: Determination of the Medial Axis

The transformed medial axis (MA) of a 2D shape, also known as the skeleton or centreline, is a set of lines that define the “geometric centre” of the shape [49,50,51,52]. The MA of the AZ is determined using the medial axis algorithm, in which a distance transform function is applied to each point of the AZ corresponding to the Euclidean distance to its nearest point on the shape boundary. By defining a threshold of the distance, we determine which points of the map are considered to be inside or outside the AZ. A common choice for the threshold value, T, is half the maximum distance. The MA is computed as the set of points equidistant from the edge of the AZ, that satisfy the following condition:(4)|d(x,y)− T| ≤ε
where epsilon is a positive value limiting the thickness of the skeleton.

The set of points satisfying the previous Equation (4) can be connected to form curves representing the MA. A navigation model based on topological and semantic relationships is then generated automatically. For this process, we used the HCMGIS add-on for QGIS [53,54], with the Skeleton/Medial Axis tool. We set the density value to 1.8 m to reduce the number of unwanted branches, using the criterion of reducing the number of vertices in the lines that represent the studied polygons and using the number of pixels of the skeleton equidistant to the two sides of each polygon [55]. This process generated a vector layer of MA lines, which represent the central skeleton of the polygon of the studied area.

At this point, places of interest were selected according to their level of importance in a pedestrian route, such as building entrances, bus stops, parking lots, pedestrian ramps, etc. For this purpose, a vectorial layer of the vertices (KP) was generated, the points were labelled, and their coordinates (X, Y, Z) were determined.

### 2.6. Stage 4: Determination and Characterisation of Feasible and Optimal Routes

The new MA line layer was divided into portions corresponding to the sections determined by the vertices of their interceptions. The total length of these portions was determined and stored in the attribute table of the MA. The vertices were also extracted as a layer of points, with heights determined by the point sampling tool of the DEM raster.

The routes were subsequently partitioned into small tracts of 3 m each, according to the minimum allowed length of a ramp. As shown in Figure 1, we also considered a threshold slope of 10%, corresponding to the minimum allowed value, to guarantee access to wheelchair users without excessive effort. Two buffers of 60 and 120 cm were established for each of these routes. The first value corresponds to the minimum width necessary for wheelchair transit, and the second corresponds to the minimum value established in Article 5 (Pedestrian routes) and Article 10 (Ramps) of Law 5/1995 “Promotion of accessibility and removal of barriers” of the Principality of Asturias [56]. The interceptions of the portions relative to the edge of the main polygon, which corresponds to the working area, were determined at the tops of each portion. Each 3 m length of track was classified according to whether or not it intercepted the main polygon at the defined buffers.

We first checked the interception, I, of the tracks with the smaller buffer (60 cm) and assigned a value of I = 0 (not passable) to the intercepting cases. We then checked the interception with the larger buffer (120 cm) for the tracks that passed the initial test, assigning a value of I = 0.5 to the tracks intercepting only the second buffer (passable tracks) and a value of I = 1 to those not intercepting any of the buffers (suitable tracks).

The vertices corresponding to the extremes of each track were determined in correspondence with the established direction of the KP vertices, and the DEM was used to determine the heights of the terrain. These data were used to calculate the gradient of each track and classify these accordingly, as flat, sloping or steep. A track was considered flat when the gradient was less than 5% (s = 1), sloping for gradients between 5% and 10% (s = 0.5), and steep for gradients greater than 10% (s = 0).

In conclusion, all of the coefficients defined and analysed above, weighting the interception with the buffers, the slope, the total width and distance, were used to classify the routes as comfortable, accessible or inconvenient for wheelchair users (Figure 4) and to determine the optimal route. In order to do so, we used a modified version of a shortest-route estimation algorithm based on randomly generated point-to-point networks [57,58], in which we introduced the aforementioned coefficients as cost values depending on the total distance.

## 3. Application of the Methodology to a Real Case

The proposed methodology was applied to a real case study involving the main access area of the Polytechnic School of Mieres (EPM) in Asturias (Spain), through which thousands of people pass every day. This public entity must be accessible to all types of people, regardless of any disability. However, although this area is the only pedestrian entrance to the building allowing wheelchair use, the surrounding area has a variety of urban obstacles to mobility. The level of deterioration of the paved area was also analysed around the access routes to determine whether any maintenance was required.

### 3.1. Initial Point Cloud and DEM by UAV Photogrammetry

A UAV flight was performed to enable generation of DEM of the test area, using a Mavic 2 Zoom UAV with a 1/2.3” CMOS 12Mpix sensor (Table 2).

This flight covered two strips at different heights to improve the convergence in the access area and the main vertical façade, and still images were acquired (see Figure 5). The model was then scaled to a long distance measured by surveying methods. Photogrammetric processing was carried out with software that uses SfM Structure from Motion techniques for image orientation and MVS Multi View Stereo for dense reconstruction (Pix4Dmapper version 4.5.6). Post-processing also included use of the basic editing tools included in this software.

The point cloud was georeferenced in Pix4Dmapper by entering distances as absolute information was not necessary. A longitudinal distance and a transverse distance measured topographically with a relative accuracy of 1 mm were included to scale the model, and two other distances were used for the validation and for the estimation of the relative accuracy. The difference in the validation distances was around 1 cm, which seemed reasonable within the scale of the survey and for the present application. The point cloud was then imported in the Cloud-Compare software to complete Stage 1.

The survey resulted in a clearly defined path of the main road and its surroundings, including the canopy and the main obstacles in the area, such as benches, lamp posts, signposts, dustbins, etc. (Figure 6). The AZ and its MA were obtained through the procedures explained in Section 2. The lines composing the MA and the 23 intercepting vertices were then determined and labelled as A, B, C, …, W. We continued with the proposed methodology by extracting the 3 m tracks and performing the analyses explained in Section 2.3, corresponding to Stage 4.

As already mentioned, wheelchair access to the EPM building is only possible through the central area, via two perimeter ramps surrounding the central stairs, corresponding to tracks EF and WU.

The bus shelter located at point A was taken as the starting point SP, and the entrance doors at points S and I were established as end points EP for the determination of possible routes.

The possible routes resulting from the proposed methodology are shown in Figure 7. The AZ, represented in light green, includes the vertices from A to I. Each track defined by the vertices inside the AZ is represented in a colour code according to their classification as comfortable (blue), transitable (yellow) or inconvenient (red). Most of the tracks were classified as comfortable (blue segments); however, inconvenient tracks (red segments) corresponding to the ramps EF and WU are unavoidable, in contrast to tracks JH and BT. The area around the bus stop (vertex A) is also an uncomfortable route as it includes tracks AB and BT and some obstacles. Several obstacles have also been included in track JH over the years, hindering transit by people with reduced mobility.

### 3.2. Route Quality Control Using Inertial Sensors

The access area of the EPM building was designed in 2004 when the University was founded. During subsequent years, there has been a significant increase in activities in the area, such as the construction of the bus shelter and other elements, which has led to a gradual deterioration of the road surface and the landscaped areas. We therefore carried out quality control of the road surface in parallel to classification of the possible routes. We used inertial sensors to determine the level of degradation of the AZ and to estimate the level/type of vibrations that wheelchair user may experience during the journey to the EMS building.

We used the International Roughness Index (IRI) [59] to analyse the roughness of the surface, determined with a VN-200 inertial sensor (Figure 8). The VN-200 sensor is a miniaturised, surface-mounted high-performance inertial navigation system (INS), which combines 3-axis accelerometers, gyroscopes and magnetometers, a barometer, a 52-channel GPS receiver, and a 32-bit processor. Moreover, it provides real-time calibrated inertial sensor measurements, by outputting high-resolution 3D drift-free position, velocity, and orientation solutions, as well as a continuous drift-free orientation solution throughout the 360 degrees of motion.

The VN-200 sensor was mounted on a wheelchair together with a 60 kg weight to simulate a disabled person (Figure 9). Other accessory instruments for real-time automation and data acquisition were also installed, including a Velodyne LiDAR, an OAK camera, a GPS and a battery. The methodology proposed in this work established the basis for further detailed analysis.

Lateral, forward and vertical displacements of the wheelchair are recorded along directions X, Y and Z, respectively (Figure 9). The V200 inertial sensor measures pitch (ΦX), roll (ΦY) and yaw (ΦZ) angles along each direction, determining the angular orientation of the wheelchair at each point of the route.

The test wheelchair was ridden along all possible routes of the AZ, and time-dependent data of the vibrational modes were obtained along the three directions. The location data reported by the GPS were used to determine the position of the wheelchair at each point of the trajectories. Time dependent angular and positional data for all the routes were collected using the commercial software of the VN-200 device, the VectorNav Sensor Explorer. The program can control the data by means of dynamic data graphs, allowing automatic visualization of the values corresponding to altitude, location, acceleration, angular velocity and metric data of the sensor. The variations in altitude represented by pitch, roll, and yaw movements are given in angular degree units (Figure 10).

A noise reduction (Python) algorithm, based on a low-frequency pass filter (after a fast Fourier transformation of the time-dependent data), was applied to the IMU measurements and their average values. The resulting deviations from the mean pitch and roll angles were analysed in order to identify deformations of the surface of each track. In the case of the yaw angle, which has a larger scale, the variation mainly corresponded to the change of direction of the measured route, and it was used for cross-validation. The results corresponding the Route 1, composed of tracks AB, BC, CD, DR, RE, EF, FQ and QI, are shown in Figure 11. The centre of the blue line corresponds to the mean value, while the width represents the standard deviation.

The main peaks in Figure 11 are expected to describe surface imperfections, which can be localised by inspecting the GPS data at the times when the peaks were raised. From these measurements, we computed the International Roughness Index (IRI) of the terrain, using Equation (5).
(5)$$IRI=\frac{\sigma \;\left(r\;*\sin\left(\Phi X\right)\right)}{L}$$
where σ corresponds to the standard deviation of the vertical displacements, *r* to the distance between the rear wheel axle and the centre of the IMU and *L* to the length of the analysed track, which in this case is 3 m.

It should be noted that only the pitch angle is included in Equation (5). We used the IRI index to classify the vertical displacements during wheelchair travel, as restless (*IRI* > 2), regular (2 > *IRI* > 1) or quiet (1 > *IRI*). A numerical convention was also established for restless (0), regular (0.5) and quiet (1).

### 3.3. Determination of the Cost Matrix of All Routes and the Optimal Route

After defining all of the numerical classifications of the properties of interest, regarding the interception of the buffers with the edge of the AZ, the slope of the tracks, and their IRI index, we integrated all of these quantities in a linear expression to define the cost of tracks, as follows.
(6)$$C_{i}=a*\;T_{i}+b\;*IRI_{i}+c\;*S_{i}\;$$
where *i* runs for all the tracks (AB, BC, CD, …) of a given route, *C_i_* is the cost value of the ith track, *T_i_* its interception, *S_i_* its slope, *IRI_i_* its *IRI* value, and *a*, *b* and *c* are weighting factors (0 < *a, b*, *c* < 1 and *a* + *b* + *c* = 1).

Equation (6) establishes a criterion for comparing each track. We then identified the routes with the best conditions in terms of slope, width, total height, and comfort. For this aim, we weighted the cost of each track by the inverse of its length, *d_i_*, and defined its efficiency, *E_i_*, per length unit as follows:(7)$$E_{i}=\frac{C_{i}}{d_{i}}$$

We then computed the total efficiency of the *jth* route and determined the optimal route using the following set of equations:(8)$$E_{tot}=\overset{nj}{\underset{i=1}{\sum }}E_{i}$$
(9)$$E_{opt}^{tot}=max_{j=1,\;\ldots ,\;N_{r}}\;$$
where *n_j_* is the number of tracks of the *jth* route, $E_{tot}\;$is the total efficiency, *N_r_* is the number of possible routes in the AZ, and $E_{opt}^{tot}\;is\;$the efficiency of the optimal route, which is the maximum of all possible routes.

## 4. Results

A database was created with all of the measurements corresponding to the characteristics of the different routes in the AZ. Figure 12 summarizes the length and the effort associated with each track in four of the fundamental routes from point A (SP) to point I (EP). The effort ranged from 0 (very difficult trajectory) to 1 (very easy trajectory). The total length of the routes ranged from 47 m to 101 m, as indicated in the title of each plot. Tracks AB, CD, and QI are common to the four routes.

## 5. Discussion

A comparison of the different methods shows that those described by [39,40,41,42] make intensive use of GIS techniques to study accessibility, while other methods, such as those described by [37,38,43,44], perform detailed analyses of accessibility inside buildings as well as in the immediate surroundings. Of the eight methods mentioned, only four deal with wheelchair users, and none of them use sensors to obtain numerical data to provide better information about the environment. Our method involves the use of GIS and photogrammetric techniques to study accessibility, primarily focused on wheelchair users. It also incorporates optical and inertial sensors, enabling a more comprehensive consideration of obstacles in urban furniture. Our method also provides up-to-date information on the condition of access routes and better documentation of architectural barriers resulting from irregularities in the road surface. This is particularly important for studying wheelchair mobility.

In the present study, the proposed methodology mainly focuses on the entrance and surroundings of a building, although it could be applied to a larger area, such as an entire city. However, analysing all buildings in a city entails handling a large volume of data. Nevertheless, our method could be used to analyse mobility in particular zones or districts of a city. Furthermore, the advantage of incorporating optical and inertial sensors, which enable better consideration of obstacles consisting of urban furniture, helps to provide more detailed information about the environment in a shorter time. This facilitates the real-time monitoring and study of urban environments.

It can be seen that the best option for the second part of the route is track BC, as track BT includes several obstacles and is also longer. As already mentioned, passage over ramps is inevitable in both tracks WU or EF, which are also the most difficult tracks as they are steep and narrow. To determine the optimal route, we used the same value, a = b = c = 1/3, for the coefficients in Equation (6) so that the three properties corresponding to intercept, slope and IRI were equally balanced. The resulting optimal route, number 1, is also the shortest. On the other hand, route number 4 was the least efficient, although it is not the longest of the possible routes, mainly due to poor conditions of the track surfaces and the presence of obstacles.

The innovative feature of this research is that it focuses on the integrated analysis of building accessibility, considering information captured by inexpensive inertial and photogrammetric sensors. Specifically, the following are innovative aspects of the proposed methodology:The workflow permits the use of diverse types of input data (point clouds from photogrammetric or laser scanning processes, architectural plans, digital 3D models, digital vector cartography, etc.);The data capture hardware is inexpensive, and the proposed processing workflow method allows rapid processing;Use of the medial axis transform algorithm (MAT) rather than a DEM enables almost automatic extraction of candidate accessible routes. This (as far as we know) is a novelty in this field;Multivariate analysis can be easily customised for optimal route selection according to user’s needs or preferences;Quantification of terrain roughness for objective assessment of route comfort is derived from IMU data and adapted to wheelchair users, which we believe is another novel aspect of this application.

## 6. Conclusions

Mobility routes are usually established during the initial design of buildings by considering different architectural criteria for defining ramps, entrance and exit widths, route width, etc. However, the route conditions may change drastically with constant use, e.g., due to deterioration of the surface or the introduction of obstacles. Analysis of how to solve these issues is usually conducted a posteriori.

General methodologies such as those developed here could be very useful for monitoring the level of accessibility to buildings and the quality of access routes. We used several technologies to describe the spatial configuration of the access zone measured by point cloud survey photogrammetry with real-time data acquisition by inertial sensors mounted on a test wheelchair to determine the inclination angles and irregularities in each section of the routes.

The proposed methodologies were put into practice in the access zone of a university building in the locality of Mieres, Spain. We were able to make a detailed report of the accessibility conditions. Moreover, we proposed an approach to find the optimal route for wheelchair users. This approach takes into consideration the width, slope and roughness of the routes. The three properties are linearly integrated into a cost function, which is weighted by the length of the tracks in order to compute the effectiveness of the routes and establish which is optimal.

We found that route 1 (ABCDREFQI) was the best. The lack of other options for accessing the building makes the comparison easier since several tracks are common to some of the routes. However, we observed some important findings; for instance, the worst route is not the longest one. Furthermore, the approach is general and applicable to other public access areas, such as schools, hospitals and public administration buildings. Ideally, this information will be made publicly available and updated in real-time so that anyone with reduced mobility would be able to consider it before deciding which route to use.

## Figures and Tables

**Figure 1 sensors-23-05491-f001:**
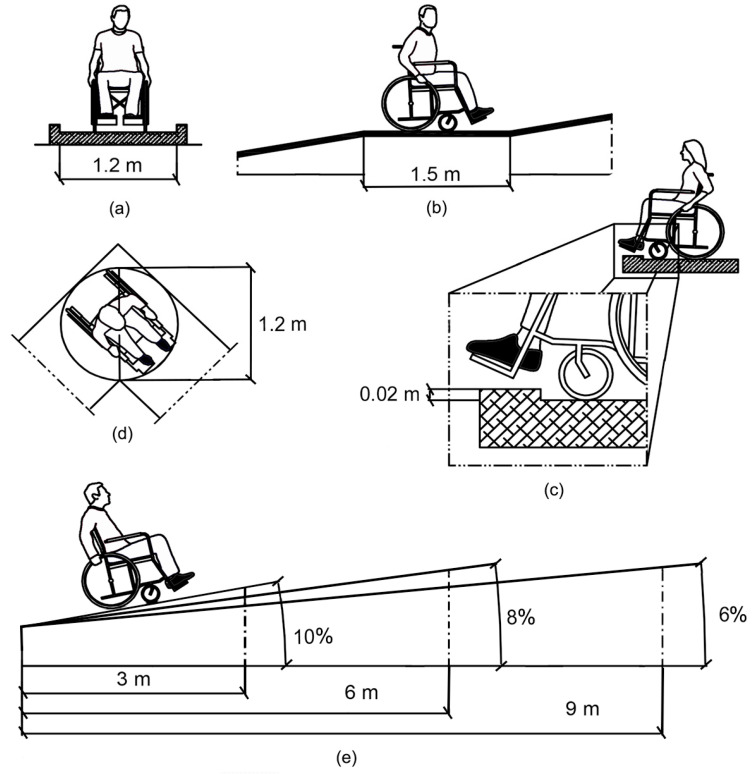
Geometric limits in the design of pedestrian ramps: (**a**) width of the platform, (**b**) minimum length of the ramp, (**c**) maximum length of the obstacle per step, (**d**) recommended width for manoeuvring, (**e**) maximum length or the ramp distances according to the percentage of slope.

**Figure 2 sensors-23-05491-f002:**
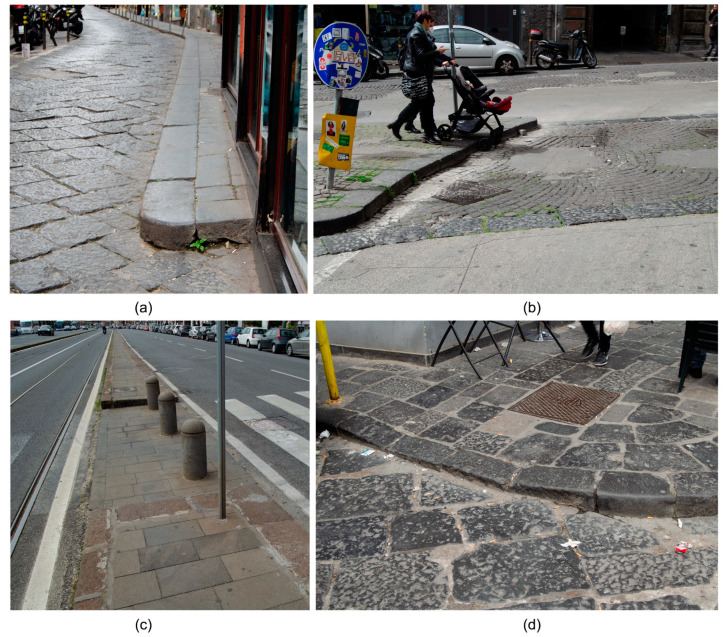
Architectural barriers in the metropolitan city of Naples: (**a**) narrow pavements, (**b**) kerbs, (**c**) obstacles at zebra crossings, (**d**) uneven level crossings.

**Figure 3 sensors-23-05491-f003:**
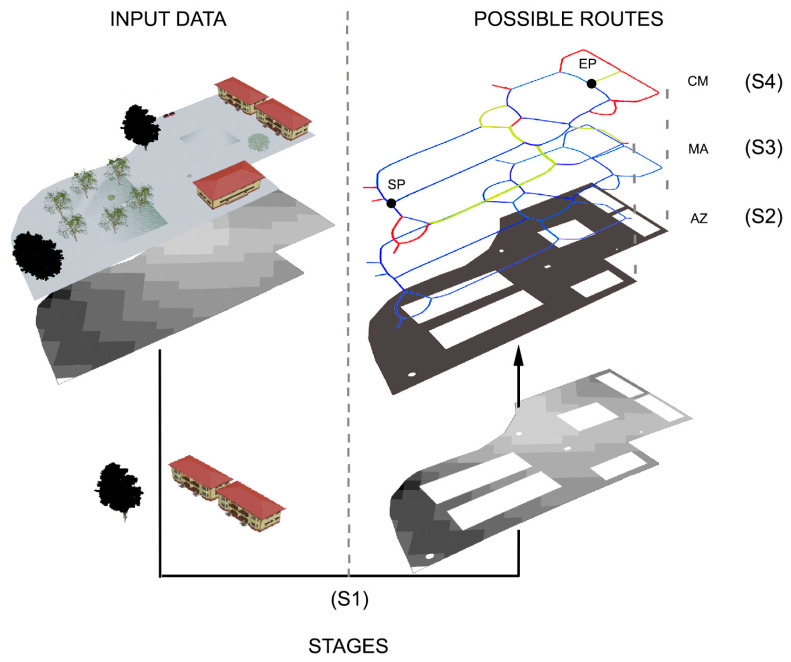
Stages S1, S2, S3 and S4 of the photogrammetric flight processes performed to obtain the accessibility zone (AZ), the medial axis (MA) and the cost matrix (CM).

**Figure 4 sensors-23-05491-f004:**
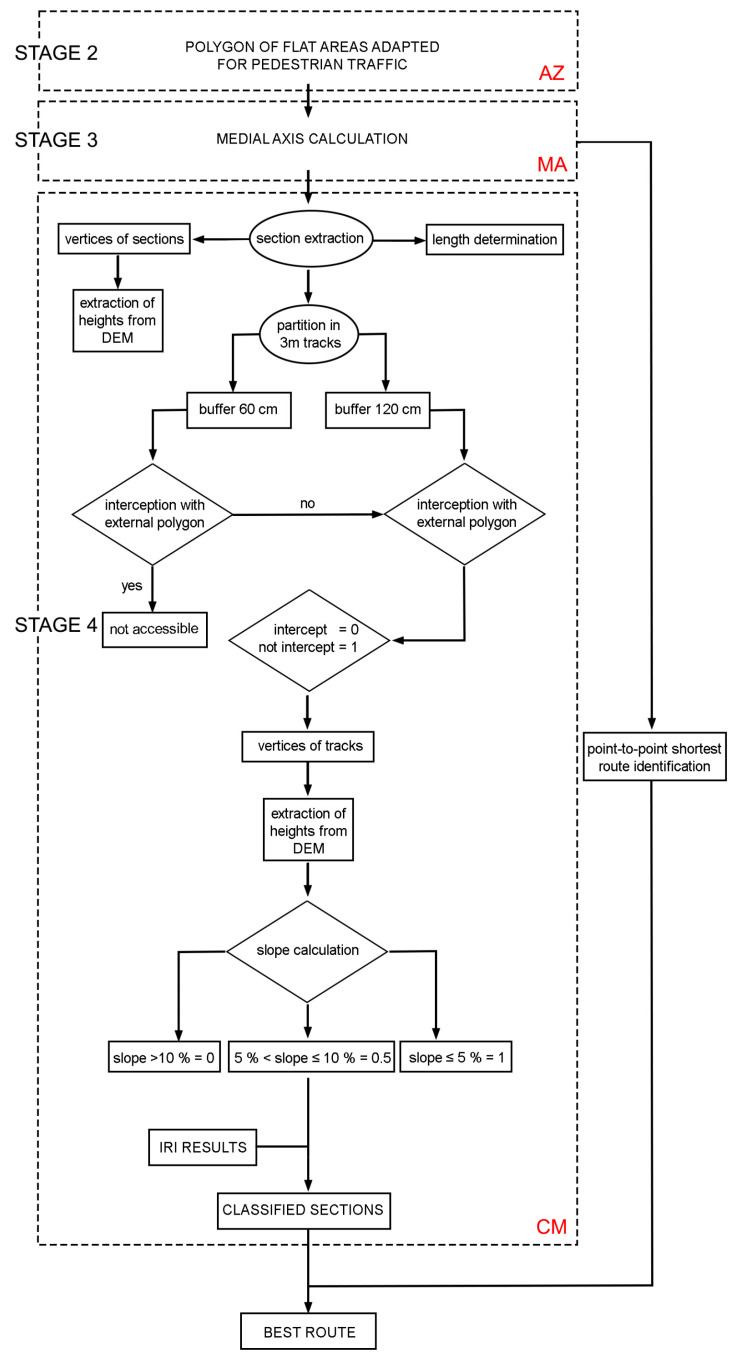
Diagram of the workflow used in Stage 4 for the generation of the CM.

**Figure 5 sensors-23-05491-f005:**
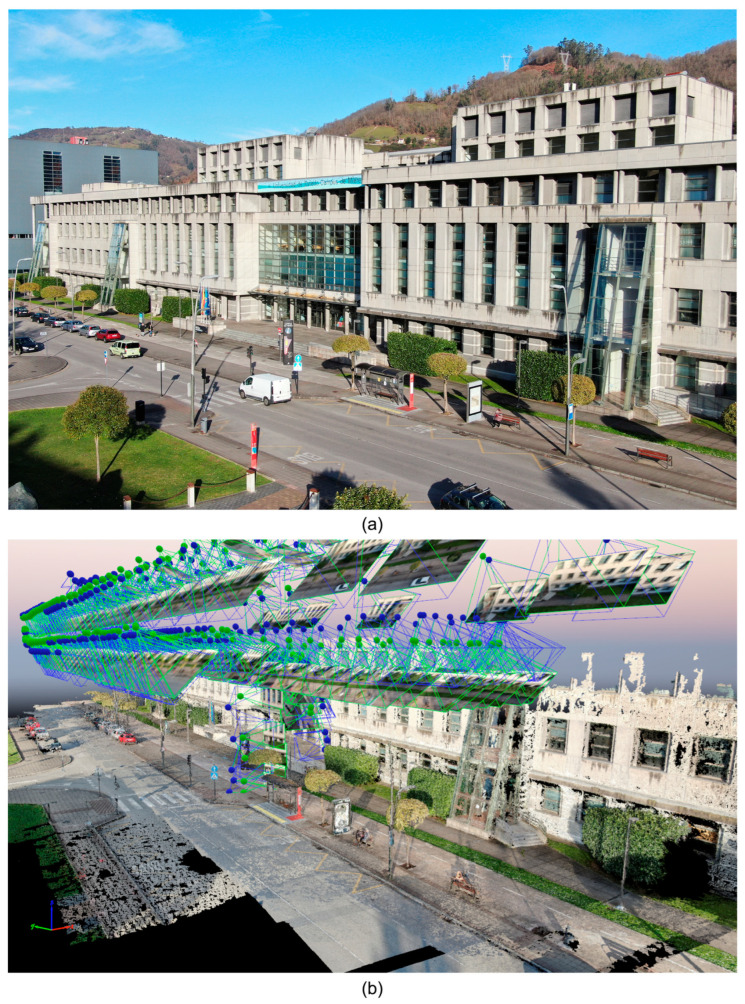
Images of (**a**) the main entrance of the EPM school and (**b**) the photogrammetric process.

**Figure 6 sensors-23-05491-f006:**
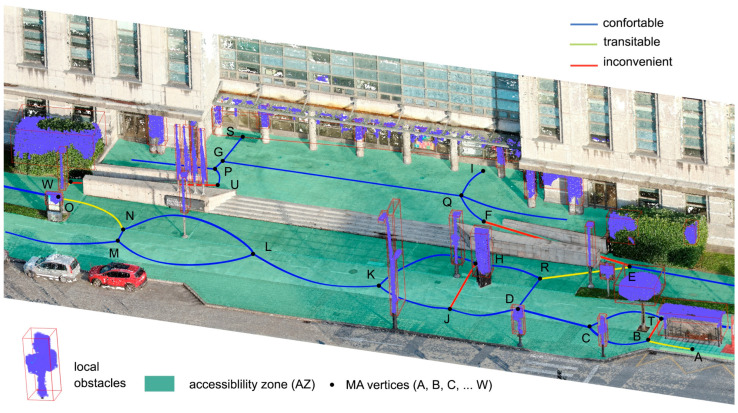
The 3D point cloud of the main area of the EPM building, with obstacles obtained during the classification in the Cloud-Compare software and its AZ and MA resulting from the proposed analysis.

**Figure 7 sensors-23-05491-f007:**
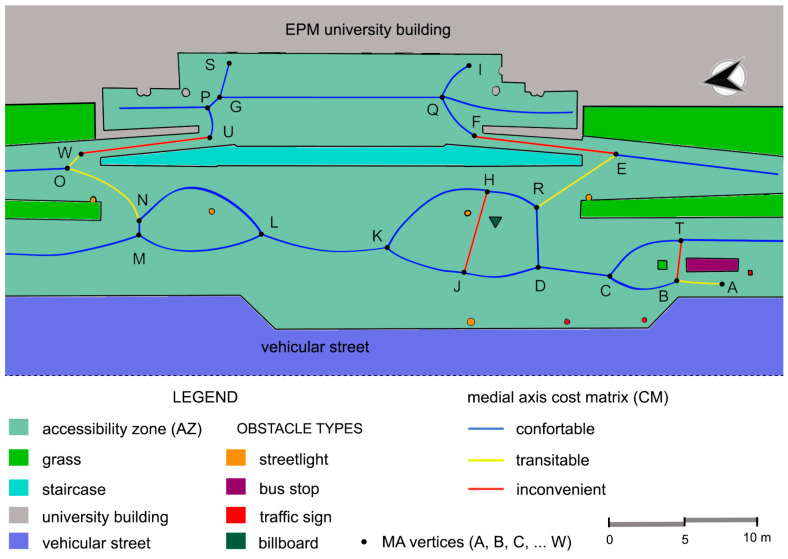
Schematic representation of the possible routes for wheelchair users to enter the EPM school.

**Figure 8 sensors-23-05491-f008:**
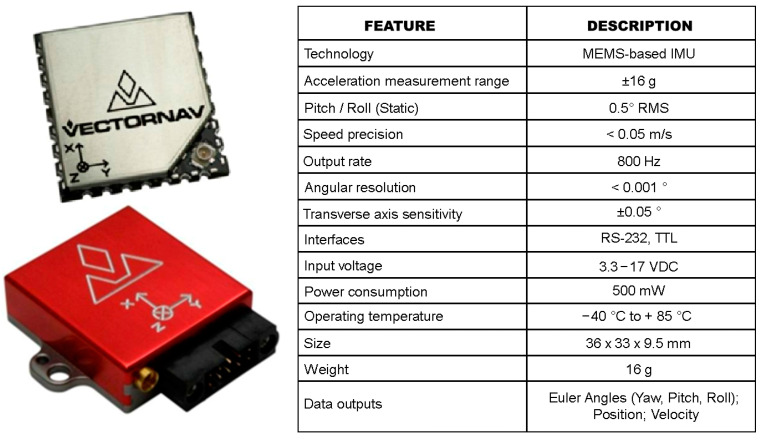
Key features of the VN-200 Inertial Navigation System (INS).

**Figure 9 sensors-23-05491-f009:**
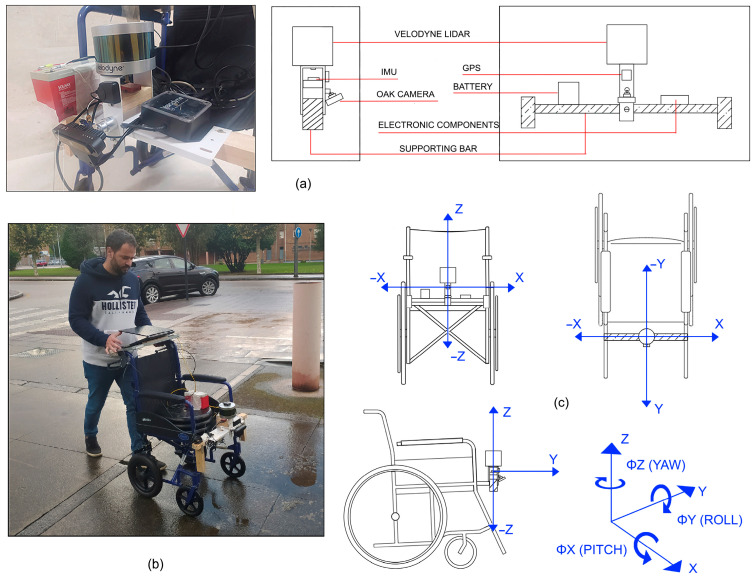
(**a**) Scheme of the used sensors: IMU inertial sensor, Velodyne LiDAR, OAK camera, GPS, and battery. (**b**) Test wheelchair with devices and sensors installed. (**c**) Schemes of different views of the wheelchair according to the coordinate system selected.

**Figure 10 sensors-23-05491-f010:**
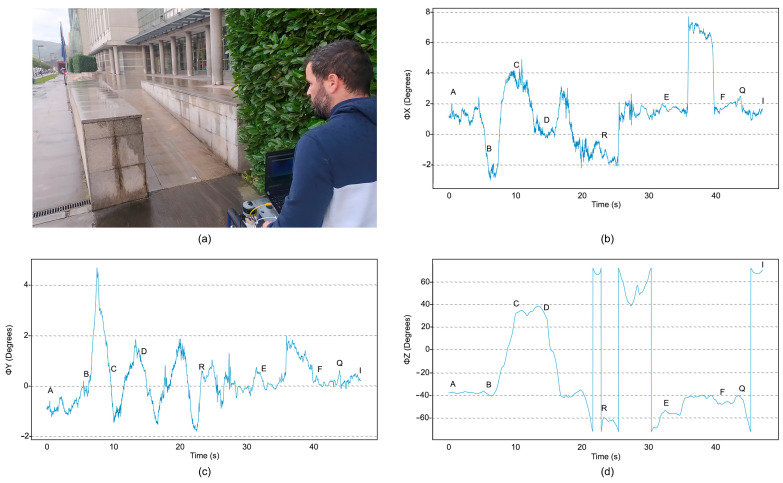
Image of the ramp to the EPM building entrance, corresponding to track EF (**a**). Panels (**b**–**d**). Dynamic data recorded by the IMU sensor during one of the runs along EF, corresponding to pitch ΦX (**b**), roll ΦY (**c**) and yaw ΦZ (**d**) angles plotted as a function of time.

**Figure 11 sensors-23-05491-f011:**
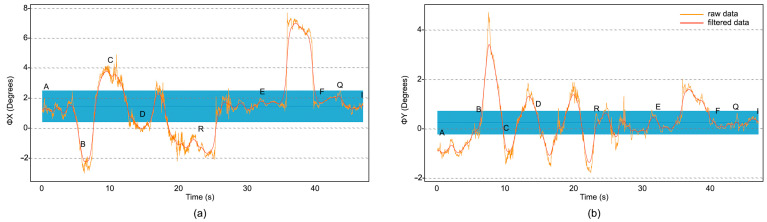
Plots of the pitch (**a**) and roll (**b**) angles as a function of time. The orange curves correspond to the raw data, the blue bars represent their mean value and standard deviation, and the (smoothed) red curves correspond to the data after applying a noise reduction algorithm based on a low frequency pass filter.

**Figure 12 sensors-23-05491-f012:**
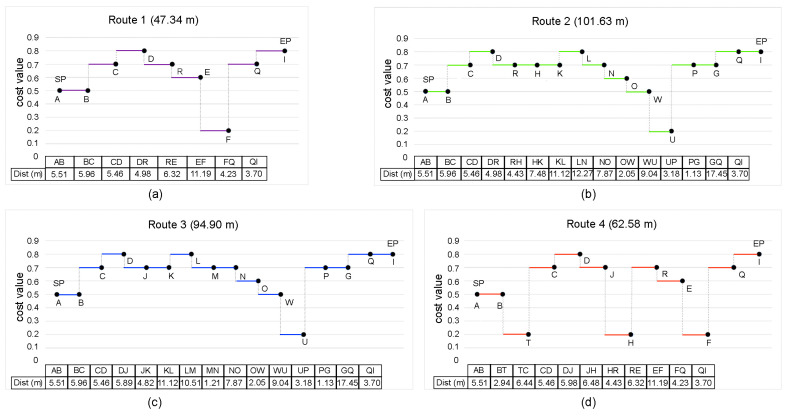
Effectiveness of the tracks corresponding to four possible routes of the AZ analysed. (**a**) Route 1: ABCDREFQI. (**b**) Route 2: ABCDRHKLNOWUPGQI. (**c**) Route 3: ABCDJKLMNOWUPGQI. (**d**) Route 4: ABTCDJHREFQI.

**Table 1 sensors-23-05491-t001:** Comparison of the proposed methodology and existing methodologies.

Reference	Accessibility Study Area	Regional (City-to-City)	Urban Area Whole City	Local Urban Area (Block)	External Area of the Building	Interior Area of the Building	Disabled (Wheelchairs)	Use of GIS Techniques	Use of Optical Sensors	Use of Inertial Sensors
[37]	Space surrounding building	✗	✗	✓	✓	✓	✗	✗	✗	✗
[38]	Space surrounding building	✗	✗	✓	✓	✓	✗	✗	✗	✗
[39]	Urban area	✗	✓	✗	✗	✗	✗	✓	✗	✗
[40]	Urban area	✗	✓	✗	✗	✗	✓	✓	✗	✗
[41]	District	✗	✓	✗	✗	✗	✓	✓	✗	✗
[42]	District	✗	✓	✗	✗	✗	✓	✓	✗	✗
[43]	Space surrounding building	✗	✗	✗	✓	✗	✗	✗	✗	✗
[44]	Space surrounding building	✗	✗	✗	✓	✗	✓	✗	✗	✗

**Table 2 sensors-23-05491-t002:** Specifications of the DJI Mavic 2 Zoom unmanned aerial vehicle.

Aircraft	
Max. ascent speed	4 m∙s^−1^ in normal mode
Max. descent speed	3 m∙s^−1^ in normal mode
Max. tilt angle	25° in normal mode and 35° in normal mode under strong wind
Operating temperature	−10 °C to 40 °C
Satellite system	GPS + GLONASS
Camera	
Sensor	1/2.3” CMOS Effective pixels: 12 MP
Max. photographic resolution	12 MP 4000 × 3000 pixels

## Data Availability

Not applicable.

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
