# Peer review of "Analysis of Building Accessibility Using Inertial and Optical Sensors"

_sensors, 2023, doi:10.3390/s23125491_

Round 1

Reviewer 1 Report

This article describes the development of a method for monitoring the accessibility and route quality of buildings, and provides experimental evidence for its effectiveness in real-world scenarios. However, the article lacks clarity in its logic and contribution, and therefore, I recommend that the article be rejected. 

1) The manuscript fails to demonstrate the novelty and contribution of the proposed method. The authors should provide a clear explanation of the innovative aspects of their approach and its main contribution.

2) The structure of the article needs to be improved, as the current organization lacks coherence. The authors should describe the organization of the manuscript in the introduction.

3) The article lacks a review of existing methods for assessing building accessibility and route quality, and does not provide a comparison with similar methods to evaluate the superiority of the proposed method.

4) The proposed method has only been validated in a small-scale experiment. Since monitoring building accessibility often requires large-scale monitoring, the authors should provide experimental evidence or discussion on the efficiency of the proposed method in larger-scale settings.

5) The Results and Discussion sections should be separated into two different headings. The authors should discuss the experimental results of their method in the Results section and provide a more detailed analysis of the method's strengths and limitations, including a comparison with other methods, in the Discussion section.

6) At Line 25, the keywords should not be numbered.

An improvement in language expression is recommended 

Author Response

See the attach PDF file

Reviewer 2 Report

The paper presents an interesting application of combining different technologies to analyze the accessibility of urban environments for wheelchair users. I would like to highlight the summary of the current state of regulations in different countries, including specific threshold parameters for evaluating accessibility. The gradual degradation of the urban environment is a real problem faced by many metropolises, and the proposed approach could be a suitable solution to improve the analysis of the current state of these zones in the future. The language level of the text seems to be appropriate, with only a few typos that are mentioned in the comments. The following comments could improve the comprehensibility of the text for readers with different specializations.

line 118 - confort - comfort

line 126 - ...Blender was use... - ... Blender was used...

line 153-164 - the described process should be explained better - it is not entirely clear, which point cloud was used in the cloud-to-mesh distance computation. Was it the cloud that was used for CSF algorithm and ground surface extraction? Wouldn't the original point cloud's real height values be more appropriate for slope determination? What is the purpose of the DEM with heights above the newly calculated CSF surface, considering that the heights would mostly be zero if the CSF was set correctly?

line 254 - SfM isn't used for "cropping and filtering of the data", it is used for the reconstruction of a 3D scene based on computer vision techniques. The relative orientation parameters of cameras in the project are determined using automatically detected and matched features in the images. There is also no detailed information about the georeferencing of the photogrammetric project - how was it done, how many ground control points (GCPs) were used and what was their distribution around the object? It is important to look at the residuals on the GCPs before analyzing the accuracy of the produced point cloud and its use for the following postprocessing in CloudCompare.

Author Response

See the attach PDF file

Round 2

Reviewer 1 Report

Thank you for addressing the concerns raised in the first round of review. However, after carefully re-evaluating the revised manuscript, I believe there are still some issues that need to be addressed before the paper can be accepted for publication. I recommend implementing the following changes:

l  Move the content of Section 1.1 to Section 2, "Related Works," and analyze the challenges or issues faced by existing methods in this section.

l  Rearrange Lines 89-97 after Line 138 and before Line 139 to highlight the superiority of the proposed method while explaining the purpose of this paper. This will enhance the logical coherence of the introduction.

l  In the discussion section, the authors should include a comparative analysis between the methodology proposed in this paper and other existing methods to demonstrate the superiority of the approach presented in this paper.

These revisions will significantly improve the quality and clarity of the manuscript. I encourage the authors to carefully consider these suggestions and make the necessary revisions accordingly.

none

Author Response

See attach file
